# Comparison of Vegetarian Sausages: Proximal Composition, Instrumental Texture, Rapid Descriptive Sensory Method and Overall Consumer Liking

**DOI:** 10.3390/foods13111733

**Published:** 2024-06-01

**Authors:** Karen P. Carhuancho-Colca, Reynaldo J. Silva-Paz, Carlos Elías-Peñafiel, Bettit K. Salvá-Ruiz, Christian R. Encina-Zelada

**Affiliations:** 1Departamento de Tecnología de Alimentos, Facultad de Industrias Alimentarias, Universidad Nacional Agraria La Molina (UNALM), Av. La Molina s/n Lima 12, Lima 15024, Peru; karencarhuancho5@gmail.com (K.P.C.-C.); celiasp@lamolina.edu.pe (C.E.-P.); 2EP Ingeniería de Industrias Alimentarias, Facultad de Ingeniería y Arquitectura, Universidad Peruana Unión, Chosica, Lima 15464, Peru; rsilva@unab.edu.pe; 3Escuela de Ingeniería en Industrias Alimentarias, Departamento de Ingeniería, Universidad Nacional de Barranca, Av. Toribio de Luzuriaga N° 376 Mz J. Urb. La Florida, Barranca 15169, Peru; 4Instituto de Investigación de Bioquímica y Biología Molecular (IIBBM), Universidad Nacional Agraria La Molina (UNALM), Av. La Molina s/n Lima 12, Lima 15024, Peru; 5Universidad Le Cordon Bleu, Av. General Salaverry 3180, Magdalena del Mar, Lima 15076, Peru; bettit.salva@ulcb.edu.pe

**Keywords:** overall liking, hedonic scale, purchase intention, flash profile, texture profile analysis, plant based sausages, sensory descriptors

## Abstract

The aim of the present research was to determine if the developed ovo−vegetarian sausage (SO), which was made with 15% chickpea flour, 51% albumin and 34% soy protein concentrate, exhibited improved physicochemical and sensory characteristics compared to vegetarian sausages available on the local market (classic vegan sausage, SC; vegan fine herb sausage, SH; and quinoa sausage, SQ). According to the physicochemical results, the developed sample, SO, presented significant differences (*p* < 0.05) compared to the others, including higher protein content, lower pH and a higher a* value. Three types of sensory analyses were conducted—flash profile, overall liking and purchase intention (to determine consumers’ willingness to purchase the product)—with the first involving 15 consumers and the second and third involving 60 participants each. Descriptors for each sample were determined using the vocabulary provided by consumers in the flash profile analysis. Descriptors for SO included ‘elastic’, ‘smell of cooked corn’, ‘characteristic flavor’, ‘pasty’, ‘soft’ and ‘pastel color’, contributing to its greater overall liking and purchase intention compared to the others. Through the hierarchical multiple factor analysis, a positive correlation was observed between the texture and sensory descriptors of the flash profile. Conversely, a correlation was found between the physicochemical characteristics (pH, a_w_, color) and overall liking and purchase intention.

## 1. Introduction

Healthy eating and concern for the environment are generating more and more awareness among consumers. For this reason, the food industry shows greater interest in the development of analogues from vegetable proteins that allow the quality characteristics of conventional meat products to be achieved, reducing environmental impact, such as greenhouse gas emissions, acidification, land and consumers’ risk of suffering from cardiovascular diseases [1,2].

The use of vegetable proteins in the preparation of meat analogues improves cooking performance, increases protein content and allows for a texture close to a conventional meat product due to its techno-functional properties such as water retention capacity, emulsion formation and gelling agents [3]. However, the total replacement of meat proteins with vegetable proteins can generate sensory changes that affect the acceptability of the product, hence the importance of a sensory analysis [4]. Additionally, among vegetarian products, there are ovo−vegetarian products, which contain, in addition to vegetable proteins, ovalbumin in their composition, which has been reported to generate a positive effect on the texture of similar sausages [5].

Sensory analysis involves determining consumer preferences and acceptability for a specific food product, providing valuable information for product development and improvement. Moreover, descriptive evaluations are also employed, where consumers (participants) describe products using terms based on sensory attributes [6]. Although, beyond appearance, consumers also condition their purchase intention with the benefits that a certain product can provide, such as its effect on health or whether the product is environmentally friendly [7]. Among the descriptive evaluations is the flash profile, which is a rapid and flexible method of sensory analysis where the consumers (participants) do not require training because they can use words from their own vocabulary to describe the product they are evaluating [8]. The purpose of this sensory analysis is to sensorially characterize the products, classify them according to the preferences and sensory attributes indicated by consumers and collect information for the development of new products [9].

Hierarchical multiple factor analysis (HMFA) belongs to the group of statistical tools used for multivariate analysis and is applied in the food industry to correlate descriptive sensory profile data with other quality characteristics of products, thereby highlighting their sensory and instrumental characteristics through configurations in the factor map [10]. It has been noted that it is generally used in the study of foods to correlate texture and color data obtained instrumentally and sensorially in products such as cheeses and yogurts, among others [11]. 

The hypothesis of our research was as follows: to obtain the optimal proportions of chickpea flour, albumin and soy protein concentrate in the elaboration of an ovo−vegetarian sausage that can achieve sensory characteristics (such as flavor, texture, aroma and appearance) equal to or superior to those of three commercially accepted sausages. Therefore, the objective of this study was to determine if the developed ovo−vegetarian sausage (referred to as SO, with 15% chickpea flour, 51% albumin and 34% soy protein concentrate) exhibited improved physicochemical and sensory characteristics compared to the vegetarian sausages already present in the local market. These sausages, which have been accepted and available for several years, are described as follows: the classic vegan sausage, SC; the vegan fine herb sausage, SH; and the quinoa sausage, SQ.

## 2. Materials and Methods

### 2.1. Preparation of Sausage Samples

Table 1 shows the characteristics and ingredients of the sausage samples evaluated. The ingredients, including egg albumin, soy protein concentrate, chickpea flour, soy protein isolate and gluten, were initially hydrated by soaking them in cold water (3 °C). Subsequently, the hydrated ingredients were blended using a food processor. The remaining ingredients were then gradually added until a uniform mixture was achieved. Oil was incorporated to form an emulsion. The mixture was then filled into synthetic casings measuring 20 mm in diameter and 12 cm in length, which were manually twisted and tied. The uncooked sausages were wrapped in polyethylene cling film and heated at 80 °C until they reached an internal temperature of 72 °C. After cooking, the sausages were rapidly cooled to 18 °C using water and stored at 4 °C for 24 h until the corresponding analyses were conducted. Furthermore, measures were taken to ensure that sausage samples purchased from the local market had the same shelf life.

Then, ovo−vegetarian sausage (SO) was compared with three other sausages available in the local market: a classic vegan sausage (SC), a vegan fine herb sausage (SH) and a quinoa sausage (SQ). These sausages were characterized based on their instrumental color, proximal composition, sensory descriptors from the flash profile, overall consumer liking, consumer purchase intention and instrumental texture analysis.

### 2.2. Proximal Composition and Colorimetric Parameters

The protein (AOAC.984.13) and fat (AOAC.2003.05) content of the SO sample was determined, in triplicate, with the methodology established by AOAC [12]. The carbohydrate content was determined by difference [13]. For commercial samples SH, SQ and SC, this information was obtained from the commercial packaging of each product.

The water activity of 2 g of sausage sample was determined using the previously calibrated AquaLab Water Activity Meter (Decagon Devices Inc.^®^, Pullman, WA, USA) [14]. The pH of homogenized sample (10 g) was determined in 100 mL of distilled water for 5 min with the use of a potentiometer (Hanna Instruments Co., Woonsocket, RI, USA) [15]. The color was determined with the Konica Minolta CR400 colorimeter, and the results were reported on the CIELab scale. Also, results were presented for the cylindrical coordinates Hue°, expressed in degrees, and Chroma [16].

### 2.3. Texture Profile Analysis (TPA)

The texture profile of the four sausage samples was determined according to the methodology proposed by Xiong et al. [17], with some modifications. Samples of 15 mm thickness were cut and conditioned at room temperature for analysis. We worked with the Brookfield Ametek^®^ (Middleboro, MA, USA) texturometer with the following configurations: a 5 kg load cell cylindrical probe plate, deformation of 50% of the original height of the sample, pre-test speed of 2 mm/s, speed constant pressure of 1 mm/s and induction force of 5 g. The attributes of hardness, cohesiveness, elasticity and chewiness were reported.

### 2.4. Sensory Analysis

#### 2.4.1. Consumers

The sensory analysis was carried out with people between 18 and 30 years old. The sample was obtained through non-probabilistic convenience sampling. For the flash profile, there were 15 consumers (gender: 40% male and 60% female), based on what was reported by Liu et al. [9] and Varela and Gaston [18], who recommended working with 15–20 consumers as it is a rapid descriptive test. For overall liking and purchase intention, a total of 60 participants were gathered (gender: 52% male and 48% female) because the UNE-EN ISO 11136:2017 [19] standard recommends working with at least 60 participants, and similar methodology was reported by Hiscock et al. [20]. 

#### 2.4.2. Flash Profile

The flash profile technique was used to generate a sensory profile for each of the sausage samples, which consisted of two stages. In the first stage, the participants tested the samples (conditioned at room temperature, with a three-digit code assigned randomly and table water used as a draft) and listed all the attributes (using their own vocabulary) that best described them. The second stage consisted of updating the personal lists of each of the participants, where a consensus was generated on some terms, and subsequently the samples were ordered in an increasing manner according to the attributes indicated in the first section [21].

#### 2.4.3. Overall Liking and Purchase Intention

Overall liking and purchase intention were evaluated with the methodology proposed by Morais et al. [22]. A total of 60 participants, aged between 18 and 30 years and comprising students and campus staff, were recruited to form the untrained consumer panel for evaluating overall liking and purchase intention. The overall liking of the optimal ovo−vegetarian sausage and three commercial vegan sausages (classic vegan sausage, vegan quinoa sausage and vegan sausage with fine herbs) was assessed using a 9-point hedonic scale (1: ‘I dislike it very much’, 9: ‘I like it very much’), while purchase intention was measured on a scale from 1 to 5 (1: ‘I would not buy it’, 5: ‘I would buy it’). Each participant evaluated the samples in a controlled environment with adequate lighting and hygienic conditions. The samples, cut to a length of 1 cm, were kept at room temperature (25 °C) and presented on disposable plates, each labeled with a unique random code. Additionally, each sample was accompanied by a glass of water for participants to cleanse their palate between tastings.

### 2.5. Statistical Analysis

The average and standard deviation of the physicochemical characteristics, texture profile, general acceptability and purchase intention of the sausage samples were quantified, and a completely randomized design was applied. An analysis of variance was performed and, if significance (*p* < 0.05) was found, the Tukey mean comparison test was performed. The flash profile was worked through a generalized Procrustes analysis, determining the PANOVA (two phase-only analysis of variance) and the sensory maps of the samples and attributes. In addition, HMFA was applied to all quantified determinations to identify correlations between the sensory profile (results of general acceptability, purchase intention and flash profile analysis) and the physicochemical characteristics and texture profile of the samples. To process the statistical data, RStudio 3.5.0 and XLSTAT 2023 programs were used.

## 3. Results and Discussion

### 3.1. Proximal Composition

Table 1 shows the protein, fat and carbohydrate content of the four sausage samples. The SH sample presented a lower protein content (4.00%), while SO presented a higher protein content (16.3%), with a significant (*p* < 0.05) value, due to the protein content of egg albumin [23], soy protein [3] and chickpea flour [24], while in the other formulations the main protein source was soy protein isolate. The difference in the protein content of the SH, SC and SQ samples can be explained by the difference in the dose of soy protein isolate that they presented. The SQ sample was the second to present a high protein content (15.6%), with a significant (*p* < 0.05) content, which can be explained by its quinoa content in addition to the soy protein isolate. On the other hand, the SO and SH samples presented lower fat content (*p* < 0.05) than the SC and SQ samples, and the SC sample had a high carbohydrate content (*p* < 0.05), while in the SH sample it was lower (*p* < 0.05). The values obtained were different from those reported by Keerthana et al. [25], who developed a vegetarian sausage with a protein content of 18.5% and fat of 3.45%.

In Figure 1, the different ovo−vegetarian sausages studied are shown. In general, sausages have an elongated cylindrical shape and a shiny surface. The SO and SC sausages have a similar structure and pale pink color. Samples SQ and SH have a brownish-yellow color with the presence of particles (fine herbs) inside.

Table 2 shows the physicochemical characteristics of the sausage samples. The SO sample presented a lower pH than the other samples (*p* < 0.05), while the SC and SQ samples presented a higher and similar pH (*p* > 0.05), which is related to the pH of the ingredients that comprise them and can be corroborated by what was reported by Kamani et al. [13] and Keerthana et al. [25], who obtained different pH values in vegetarian sausages developed with different sources of vegetable protein and at different proportions. On the other hand, the SO sample presented greater a_w_ (*p* < 0.05) than the other samples. The results obtained are within the water activity range of conventional sausages [26].

### 3.2. Colorimetric Parameters

Table 2 shows that the SO sample presented lower luminosity (L*), a greater tendency towards red (+a*) and a lower tendency towards yellow color (−b*) compared to the other sausages (*p* < 0.05) and SC presented an a* value near to that of SO, while SQ presented greater L* and SH less a* (*p* < 0.05). In other research, L* values of 68.68–70.46 [27] and 59.36–68.48 [28] for conventional sausages were reported, which allow corroboration of the results obtained.

The SO and SC samples presented a greater red hue than that reported by Keerthana et al. [25] for a vegetarian sausage. The SQ sample presented a yellow hue greater than the values of 15.00–23.40 [29] and 7.32–8.32 [30] of conventional sausages obtained in other investigations, while SO, SC and SH presented a yellow tone close to these.

There were significant differences (*p* < 0.05) between the Hue° and Chroma values of the four sausage samples. According to the results, the SO sample presents a pale pink color, the SC sample has a pinkish tone that is closer to red and less pale, the SQ sample a yellow and brighter tone and sample SH an even yellower shade than SQ but pale like SC. These values were different from the Hue° (66.6°) and Chroma (13.7) of a chicken sausage [28]. The differences in the color of the sausage samples are related to the different ingredients used in their production and at different doses [9].

### 3.3. Texture Profile Analysis (TPA)

Table 2 shows the results of the texture profile analysis of the sausage samples. There were no significant differences (*p* > 0.05) in the hardness of the SH (19.74 N) and SO (19.51 N) samples, and it was higher than in the other samples due to the gelation capacity of soy protein [3] and egg albumin [5]. Chickpea flour also influenced the hardness of the SO sample, due to its relatively high gelation capacity and its high fiber content that allows water absorption and contributes to hardness [31]. In addition, the starch content of chickpea flour generated an increase in hardness and chewiness due to the swelling of the starch granules embedded in the protein matrix [32]. The SC sample presented a hardness similar to that of a vegetarian sausage (13.4 N) reported by Kamani et al. [15]. The SO and SH samples also presented greater chewiness (*p* > 0.05) and SC presented less chewiness than the other samples (*p* < 0.05). However, the SO, SQ and SH samples presented greater hardness and chewiness than a chicken sausage with values of 16.7 N and 6.7 N, respectively [33]. Samples SO, SQ and SH presented similar (*p* > 0.05) and greater cohesiveness than sample SC, indicating that these samples better preserved their integrity during cooking due to the forces of molecular bonds within the product [34]. Zhao et al. [5] reported the positive effect of egg albumin on the cohesiveness of vegetarian sausages, which corroborates the result obtained. There were no significant differences between the elasticity values of the four samples (*p* > 0.05). The results were higher than the elasticity of a chicken sausage with 30% hemp flour (0.79) but lower than that of a conventional chicken sausage, with a value of 0.91 [33]. The SC and SQ samples presented a similar value of adhesiveness (*p* > 0.05) which was higher than the other samples, which may be related to the moisture content of the product or to a greater presence of dietary fiber in its composition that improves the retention of water and consequently increases adhesiveness [33].

**Table 2 foods-13-01733-t002:** Mean and deviation of the physicochemical characteristics and texture profile of the ovo−vegetarian (SO), classic vegan (SC), quinoa vegan (SQ) and fine herb vegan (SH) sausage.

Sample	pH	a_w_	L*	a*	b*	Hue°	Chroma	Hardness	Cohesiveness	Elasticity	Adhesiveness (N-mm)	Chewiness (N)
SO	7.34 ± 0.01 ^a^	0.990 ± 0.001 ^c^	65.5± 0.17 ^a^	17.9 ± 0.05 ^d^	7.27 ± 0.08 ^a^	22.2 ± 0.28 ^a^	19.3 ± 0.02 ^a^	19.5 ± 0.90 ^c^	0.63 ± 0.06 ^b^	0.83 ± 0.06 ^a^	0.32 ± 0.00 ^a^	10.3 ± 0.86 ^c^
SC	7.52 ± 0.01 ^c^	0.986 ± 0.001 ^b^	66.6 ± 0.02 ^c^	17.3 ± 0.03 ^c^	16.6 ± 0.04 ^b^	43.9 ± 0.05 ^b^	24.0 ± 0.05 ^b^	13.4 ± 0.21 ^a^	0.40 ± 0.00 ^a^	0.80 ± 0.00 ^a^	0.40 ± 0.03 ^b^	4.29 ± 0.07 ^a^
SQ	7.51 ± 0.01 ^c^	0.986 ± 0.001 ^b^	66.9 ± 0.03 ^d^	7.87 ± 0.11 ^b^	42.8 ± 0.17 ^d^	79.6 ± 0.12 ^c^	43.5 ± 0.18 ^d^	17.9 ± 0.81 ^b^	0.57 ± 0.06 ^b^	0.80 ± 0.00 ^a^	0.40 ± 0.00 ^b^	8.07 ± 0.48 ^b^
SH	7.38 ± 0.01 ^b^	0.984 ± 0.001 ^a^	66.1 ± 0.06 ^b^	4.09 ± 0.07 ^a^	26.8 ± 0.03 ^c^	81.3 ± 0.15 ^d^	27.2 ± 0.02 ^c^	19.7 ± 0.10 ^c^	0.60 ± 0.00 ^b^	0.83 ± 0.06 ^a^	0.37 ± 0.03 ^ab^	9.86 ± 0.63 ^c^

Different letters indicate that data in the same column have significant statistical differences (*p* < 0.05).

### 3.4. Sensory Analysis

#### 3.4.1. Flash Profile

Figure 2 presents the two-dimensional consensus space, where the four evaluated samples are located. According to Rodríguez-Noriega et al. [35] and Lozano et al. [36], it is necessary to work with two dimensions if they explain more than 80% of the variance of the data. For this reason, it was considered sufficient to analyze the flash profile data of the sausages with the first and second dimensions. In Figure 2, it can also be seen that consumers considered three groups of samples: the first group was the SH and SQ samples and the second and third groups were the SC and SO samples, respectively. According to the descriptors seen in Figure 3, the SC sample was characterized as soft, pale in color, characteristic odor, characteristic flavor, porous and flexible; samples SH and SQ were described as having a bean flavor, spices, pale and soft color.

The SO sample was characterized by being elastic, smelling of cooked corn, strong color smell, characteristic flavor, pasty, soft and pastel color. The attributes of characteristic flavor and smell, salty, soft texture and pale color have been used as descriptors in a Bologna sausage, where it is mentioned that they are terms frequently used to describe this type of meat product [14,37]. Cruz-López et al. [38] also reported descriptors of grassy smell, porous, pasty, grassy taste, plastic and stretchy in their powdered cricket-based sausages.

#### 3.4.2. Overall Liking and Purchase Intention

In Figure 3, no significant differences (*p* > 0.05) are observed in overall liking and purchase intention between the SH and SQ samples. The analysis was conducted with students who adhere to a vegetarian diet. The SO sample presented higher overall liking (5.65/9.00) and purchase intention (3.43/5.00) than the other samples. The acceptability of the product by the consumer is conditioned by aspects such as its appearance, texture, color and flavor [6], which were better in SO according to consumer preference in the present study. In the case of meat products, consumers prefer a color that gives the sensation of meat; this explains that the SO and SC samples presented greater acceptability and purchase intention due to their more red-pink hue than the SH and SQ samples, which presented a yellow hue [22]. The use of a high dose of vegetable protein and fiber can affect the acceptability of the product by giving it a more pasty or floury consistency [6,22], so the low acceptability may be related to the ingredients used in the preparation and the proportions. Morais et al. [22] developed a collared peccary sausage that presented an overall liking of 7.84 and a purchase intention of 4.19—values higher than what was obtained in our present study. Mazumder et al. [39] reported a general acceptability of 5.70 for a vegetarian sausage based on oyster mushroom, chickpea flour, gluten and pea protein isolate, which is also higher than the value obtained in our present study.

### 3.5. Hierarchical Multiple Factor Analysis (HMFA)

The results of the HMFA are shown in Figure 4. The results of the analysis were presented in two dimensions because they explained 86.86% of the variance of the data. Sensory attributes from the flash profile analysis (descriptive sensory) were positively correlated with texture. Likewise, there was a correlation between the physicochemical characteristics (pH and a_w_), color and the results of overall liking and purchase intention (hedonic); however, there was no correlation between the sensory descriptors and texture and the other variables, as shown in Figure 4a. It has been reported that descriptive sensory data correlate with texture because it is one of the attributes that consumers define most in tasting. However, it is also linked to the overall liking of the product since, in the case of sausages, descriptors such as hardness, chewiness and saltiness are those that lead to greater overall liking and purchase intention [6].

The lack of correlation between descriptive and hedonic sensory data can be explained by the discrepancy between consumers from the flash profile analysis that made it difficult to define consumer preferences towards a product. Valli et al. [40] also obtained a positive correlation between sensory analysis data and the texture of a salami. On the other hand, there is a correlation between color and overall liking and purchase intention, since color is part of the appearance and the consumer generally expects a product to present the same color as the raw material from which it comes [22]. The correlation between pH and water activity with hedonic data can be explained by the high overall liking that the SO sample had with a lower pH and higher a_w_ than the other samples and how the acceptability decreased with an increase in pH and a decrease in water activity. No such information has been found in sausage studies, but it has been reported that pH significantly influences (*p* < 0.05) the acceptability of drinks with red cabbage and hibiscus extracts due to the acidity that can increase or decrease with different doses of the ingredients in the formulation [41].

There was dispersion in the color results for the SO sample (Figure 4b), unlike the physicochemical, texture, hedonic and descriptive sensory results where everyone responded similarly. In the SC sample, there was greater dispersion in all response variables; in the SH sample, greater consensus was obtained for all response variables, especially in the physicochemical and texture results; and in the SQ sample, a small difference was obtained between the response variables, but a homogeneous consensus was reached, unlike SC and SO where a greater difference was obtained. The dispersion of the results of the physicochemical properties, color and texture can be explained by factors that affect the evaluation, the margin of error of the equipment or variations in the different sample points and the variations in the descriptive and hedonic sensory results due to the different vocabulary used by the consumer and the difference in preference, respectively [38,41].

The map of individual factors (Figure 4c) shows the confidence ellipses to determine the existence of significant differences (*p* < 0.05) between the sausage samples. The SQ and SH samples presented similar characteristics (*p* > 0.05) due to the descriptors correlated with dimensions 1 and 2. However, the SO and SC samples presented the opposite result regarding dimensions 1 and 2 (*p* < 0.05). Samples SQ and SH are similar to SC with respect to dimension 1, but are the opposite with respect to dimension 2, while SQ and SH are similar to SO in dimension 2, but different with respect to dimension 1. According to Figure 4d, the SO sample presented greater general acceptability and purchase intention, greater hardness, elasticity, a_w_ and a* value than the other samples. The SH and SQ samples presented greater hardness, cohesiveness and chewiness, after SO, and high values of Hue°, Chroma and b*; SC presented higher luminosity and pH than the other samples. The SH, SQ and SC samples presented greater adhesiveness than SO.

## 4. Conclusions

The ovo−vegetarian sausage presented a better sensory profile and physicochemical characteristics than commercial sausages. There were differences in the instrumental color (*p* < 0.05) among the four sausage samples, and the ovo−vegetarian sausage and the vegan fine herb sausage presented similar values (*p* > 0.05) for hardness and chewiness. Consumers characterized the ovo−vegetarian sausage with different descriptors from those they used for the commercial samples: smell of cooked corn, strong smell, soft, elastic, pasty, and pink color. Moreover, the ovo−vegetarian sausage presented greater flavor and overall purchase intention than the other samples, with values of 5.65/9.00 and 3.45/5.00, respectively.

The results of the hierarchical multiple factor analysis (HMFA) showed a positive correlation between texture and the results of the flash profile analysis, but a low correlation with the physicochemical variables, color, and hedonic characteristics. In summary, the development of this new ovo−vegetarian sausage may provide a healthy and tasty alternative for those looking to reduce their meat consumption or follow an ovo−vegetarian diet. Additionally, it offers an opportunity to innovate in the food market and meet the growing demand for more sustainable food options.

As future research, additional studies will be conducted to produce emulsified meat products by incorporating coloring-antioxidant compounds, derived from agro-industrial waste such as fruit peels using ultrasound technology, and assessing their impact on sensory attributes.

## Figures and Tables

**Figure 1 foods-13-01733-f001:**
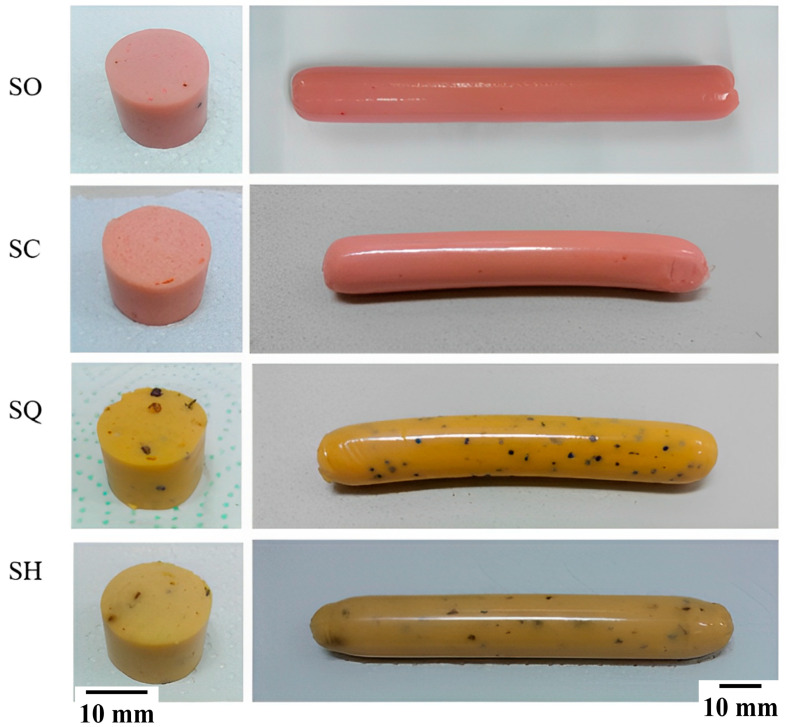
Ovo−vegetarian (SO), classic vegan (SC), quinoa vegan (SQ), and fine herb vegan (SH) sausage samples. Scale bars: 10 mm.

**Figure 2 foods-13-01733-f002:**
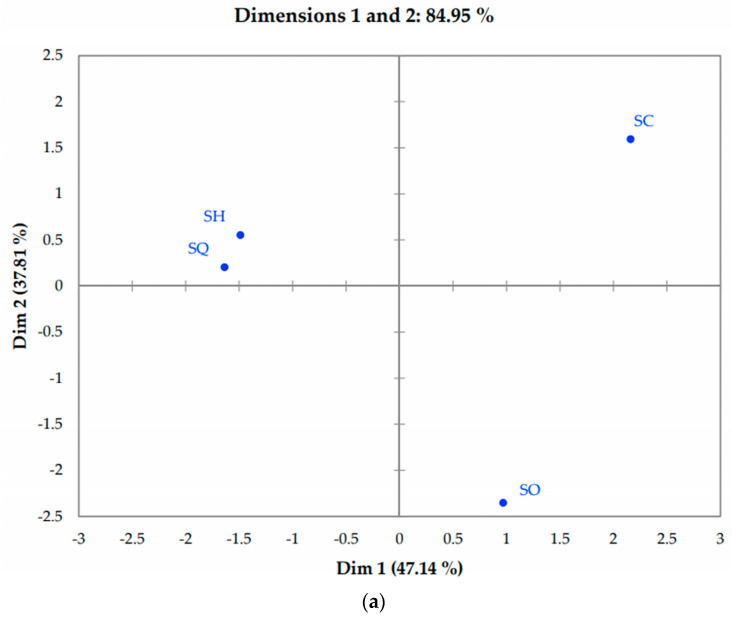
Two-dimensional graph using the flash profile: samples (**a**) and attributes (**b**). SO: ovo−vegetarian sausage; SH: vegan fine herb sausage; SQ: vegan quinoa sausage; SC: classic vegan sausage.

**Figure 3 foods-13-01733-f003:**
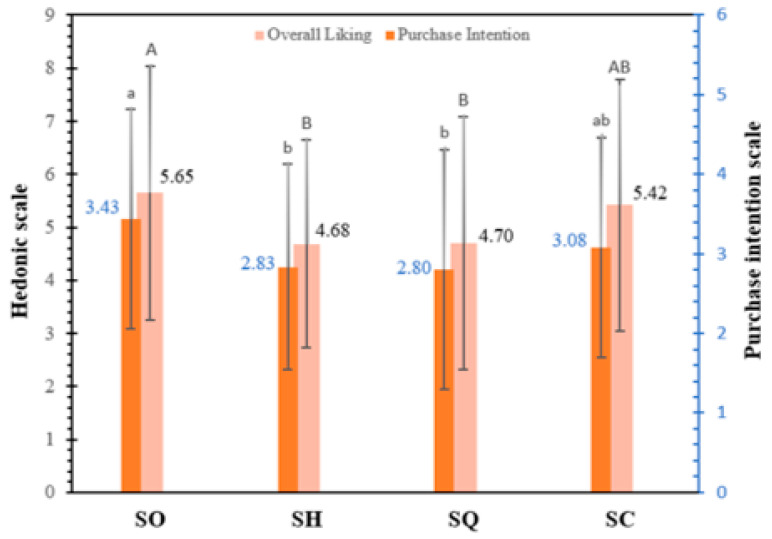
Evaluation results of overall liking (9-point hedonic scale) and purchase intention (5-point scale) of the ovo−vegetarian sausage (SO) and three commercial vegan sausages: SH (vegan fine herb sausage), SQ (vegan quinoa sausage) and SC (classic vegan sausage). Different uppercase letters indicate that data in the overall liking have significant statistical differences (*p* < 0.05). Different lowercase letters indicate that data in the purchase intention have significant statistical differences (*p* < 0.05).

**Figure 4 foods-13-01733-f004:**
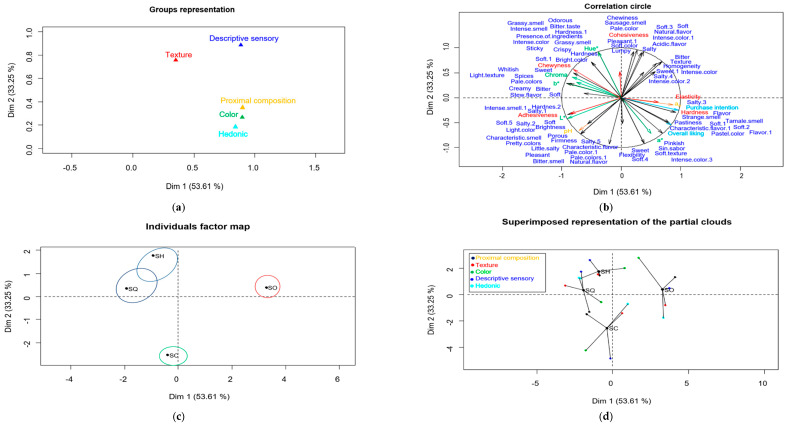
Hierarchical multiple factor analysis (HMFA) generated with the sensory profile and the physicochemical characteristics and texture profile of the sausage samples: (**a**) two-dimensional graphic representation of the response variables evaluated in the sausage samples, (**b**) correlation of the sensory profile with the physicochemical characteristics and texture profile, (**c**) confidence ellipses of the sausage samples with a confidence level of 95% (1 − α = 0.95), (**d**) comparison and statistical approximation of the sausage samples.

**Table 1 foods-13-01733-t001:** Proximate composition of ovo−vegetarian (SO), classic vegan (SC), quinoa vegan (SQ) and fine herb vegan (SH) sausage.

Sample	Ingredients	Protein (%)	Fat (%)	Carbohydrates (%)
SO	Egg albumin, soy protein concentrate, chickpea flour, soy protein isolate, wheat gluten, vegetable oil, salt, garlic powder, onion powder, oregano, sausage flavoring, beet extract and carmine.	16.3 ^a^	2.57 ^c^	6.68 ^b^
SC	Soy, soy protein, vegetable oil, potato starch, garlic, onion, flavoring, annatto, smoke essence, natural gum and salt.	14.7 ^c^	6.00 ^b^	7.80 ^a^
SQ	Soy, isolated soy protein, vegetable oil, potato starch, garlic, onion, flavoring, smoke essence, annatto, aromatic herbs, red quinoa, black quinoa, natural gum and salt.	15.6 ^b^	6.30 ^a^	4.20 ^c^
SH	Soy, isolated soy protein, vegetable oil, potato starch, garlic, onion, flavorings, smoke essence, annatto, aromatic herbs, fine herbs, thyme, rosemary, natural gum and salt.	4.00 ^d^	2.30 ^d^	2.30 ^d^

Different letters indicate that data in the same column have significant statistical differences (*p* < 0.05).

## Data Availability

The data presented in this study are available upon request from the corresponding author. The data are not publicly available due to privacy.

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
