# Peer review of "Comparison of Vegetarian Sausages: Proximal Composition, Instrumental Texture, Rapid Descriptive Sensory Method and Overall Consumer Liking"

_foods, 2024, doi:10.3390/foods13111733_

Round 1
Reviewer 1 Report
Comments and Suggestions for Authors
Authors performed different sensory tests (flash profile with 15 panelists, overall liking and purchase intention with 60 consumers) but they did not provide ethics committee approval. The informed consent and personal data protection are important documents that have to be accepted by assessors before to start the sensory tests.
Please, explain how the food safety aspect of the SO sausages prepared by the researchers then sensory tested was assured (if and how the possible undesirable microbial growths during storage was controlled and verified).
Table 1: Authors should explain why they did not use for the formulation of SO sample garlic, onion, flavoring, annatto, smoke essence as done for preparation of SC, SQ and SH.
3.3.1 Flash profile and Figure 2: I think that "strange smell" and "pretty colors" can not be considered useful sensory descriptors for the flash profile analysis. What means "smelling like a tamale"?
Author Response
Response to Reviewer 1 Comments
Response: First of all, thank you very much for your appreciation of our manuscript by the reviewer. In “blue font”, you will find our answers to your suggestions/queries.
- Authors performed different sensory tests (flash profile with 15 panelists, overall liking and purchase intention with 60 consumers) but they did not provide ethics committee approval. The informed consent and personal data protection are important documents that have to be accepted by assessors before to start the sensory tests.
Response: We appreciate your observation. As you point out, informed consent was requested prior to sensory analysis for consumer participation. The project was approved by the Ethics Committee of the “Universidad Peruana Unión” from Lima-Peru, with the approval code “N°2022-CE/FIA-0016” in November 14th, 2022. Moreover, the document was sent to Mr. Dusan Duric, MDPI Assistant Editor.
- Please, explain how the food safety aspect of the SO sausages prepared by the researchers then sensory tested was assured (if and how the possible undesirable microbial growths during storage was controlled and verified).
Response: Thank you for your comment. Regarding concerns about possible undesirable microbial growth in the processed sausage samples (SO), we assure you that quality was maintained. It's important to note that stringent measures were in place to ensure safety throughout the entire process, including ingredient sourcing, manufacturing, storage, and sensory analysis. Ingredients were sourced from companies adhering to Good Manufacturing Practices (GMP) and Hazard Analysis and Critical Control Points (HACCP) standards. Manufacturing procedures followed strict protocols, including the use of protective gear such as gloves, masks, and hair nets, as well as ensuring surfaces and utensils were thoroughly disinfected. Moreover, during the sausage production process, adherence to Good Manufacturing Practices was rigorously maintained, and the sausages were stored under refrigeration for only one day. Prior to production, microbial growth was controlled, and physicochemical parameters such as pH, water activity, and color were regularly monitored and found to be within acceptable ranges. Furthermore, hygienic conditions were strictly upheld during all sensory tests.
- Table 1: Authors should explain why they did not use for the formulation of SO sample garlic, onion, flavoring, annatto, smoke essence as done for preparation of SC, SQ and SH.
Response: Thanks for your question. In this regard, we indicate that we did use them in the production process, but they were not detailed, they were only indicated as condiments. We will detail it in the manuscript. The updated information is in Table 1.
- 3.1 Flash profile and Figure 2: I think that "strange smell" and "pretty colors" can not be considered useful sensory descriptors for the flash profile analysis. What means "smelling like a tamale"?
Response: Done. We appreciate your feedback. Consequently, the words have been corrected as per your suggestions. The descriptor “strange smell” has been replaced with “strong color”, and “pretty color” with “pale pink color”.
Additionally, “tamale” refers to a traditional Peruvian dish made with corn (Zea mays). In this context, the expression “smelling like a tamale” indicates that a group of consumers who participated in the sensory evaluation associated the smell of the sausage with the aroma of a tamale. We recognize that this term may not be universally recognized, so “smelling like a tamale” has been changed to “smelling like cooked corn”. This adjustment aims to prevent confusion for readers unfamiliar with the term 'tamale'.
Reviewer 2 Report
Comments and Suggestions for Authors
Title: I suggest changing the title and instead of Descriptive Sensory Profile use Rapid Descriptive Sensory Method
Abstract: The abstract is very confusing and repetitive. The objective of the study is not indicated. It is not indicated which one of the sausages was developed by the researchers. It is not indicated that the flash profile and the overall liking and purchase intention were carried out with two different groups of consumers. “Descriptive profile” is used as a synonym for “Flash profile” and that is not correct. Between which physicochemical properties were significant differences found? Information is missing in the abstract.
L50-52. Sensory analysis is not carried out only with that objective. This phrase gives rise to confusion to the reader who is not familiar with this scientific discipline.
L54-56. In this sentence you are mixing descriptive techniques that must be performed with panels of highly trained judges with techniques that can be performed with semi-trained judges or consumers. There is no discussion of rapid descriptive methods or why they were developed, or the advantages they have over classical descriptive methods. More information should be given about the flash profile technique.
L67-70. If the objective was only to characterize 4 vegetarian or vegan sausages from a physicochemical and sensory point of view, the work does not present anything new. If the goal was to design an ovo-vegetarian sausage and compare it with others on the market, it could be more interesting. I suggest rewriting the objective and the paper.
L73-79. The preparation of the ovo-vegetarian sausage is poorly explained. How was that formulation developed? The most interesting part of the work could be the development of this product and its comparison with others on the market. Was it considered when buying sausages on the market that they had a similar shelf life?
L102-105. There is a lack of bibliographic references and information on consumer recruitment. The term “untrained panelists” would seem to be synonymous with consumer, but this is not clear and leads to confusion for the reader. It would also seem that there were two recruitments: one of individuals to carry out the flash profile methodology and another to evaluate acceptability and purchase intention of the samples, but this is not well explained. The number of consumers to evaluate acceptability and purchase intention is very low. Were the recruited individuals regular consumers of vegetarian products? This information is very important. Why wasn't a quick descriptive methodology carried out with the same consumers who evaluated acceptability, such as CATA or RATA? The results could have been more reliable.
L112-114. What were the terms that were agreed upon?
L117-122. There is a lot of information missing about how this test was performed. Where was the study carried out? Were the samples presented at random? What was the serving temperature? Was there any prior preparation of the samples?
L129. ANOVA on flash profile data? This is a mistake.
L245-259. The sample with the highest overall liking did not reach the minimum commercial acceptability (6.0 on a 9-point scale). Could it have been because the consumers were not vegetarians? Everything is very poorly explained and there is no good discussion of the results. If a larger number of consumers had been used for the study, groups with different preference patterns could have been found, which would have enriched the results.
L269-L270. A figure is not necessary to present. A table with the average values and their standard deviations would be more useful to the reader. Furthermore, the figure leads to confusion by presenting data from two different scales.
Author Response
Response to Reviewer 2 Comments
Response: First of all, thank you very much for your appreciation of our manuscript by the reviewer. In “blue font”, you will find our answers to your major suggestions/queries.
- Title: I suggest changing the title and instead of Descriptive Sensory Profile use Rapid Descriptive Sensory Method
Response: Done. We consider your suggestion and we modified the manuscript title by: “Comparison of Vegetarian Sausages: Proximal Composition, Instrumental Texture, Rapid Descriptive Sensory Method and Overall Consumer Liking”.
- Abstract: The abstract is very confusing and repetitive. The objective of the study is not indicated. It is not indicated which one of the sausages was developed by the researchers. It is not indicated that the flash profile and the overall liking and purchase intention were carried out with two different groups of consumers. “Descriptive profile” is used as a synonym for “Flash profile” and that is not correct. Between which physicochemical properties were significant differences found? Information is missing in the abstract.
Response: Done. We appreciate your comment and, we consider it valuable, for improving the quality of the manuscript. The summary has a limit of 200 words, which means we aim to provide substantial and concise information in the first draft of the manuscript. However, we have proceeded to modify it based on your valuable suggestions, incorporating the following information in the abstract section:
“The aim of the study was to determine if the Ovo-vegetarian Sausage, developed as SO (with 15% chickpea flour, 51% albumin, and 34% soy protein concentrate), exhibited improved physicochemical and sensory characteristics compared to vegetarian sausages available on the local market (Classic Vegan Sausage SC, Vegan Fine Herb Sausage SH, and Quinoa Sausage SQ). According to the physicochemical results, the developed sample SO presented significant differences (p < 0.05) compared to the others, including higher protein content, lower pH, and a higher a* value. Three types of sensory analyses were conducted: the Flash Profile, Overall Liking, and Purchase Intention (to determine consumers' willingness to purchase the product), with the first involving 15 panelists and the second and third involving 60 panelists each. Descriptors for each sample were determined using the vocabulary provided by consumers in the Flash Profile analysis. Descriptors for SO included 'elastic', 'smell of cooked corn', 'characteristic flavor', 'pasty', 'soft', and 'pastel color', contributing to its greater overall liking and purchase intention compared to the others. Through the HMFA analysis, a positive correlation was observed between the texture and descriptive sensory of the flash profile. Conversely, a correlation was found between the physicochemical characteristics (pH, aw, color) and overall liking and purchase intention.”
- L50-52. Sensory analysis is not carried out only with that objective. This phrase gives rise to confusion to the reader who is not familiar with this scientific discipline.
Response: Thanks for your comment and acknowledge that the intended purpose of sensory analysis in the study was not accurately expressed. Therefore, the following modification is proposed in lines 97-100, in the improved manuscript:
“Sensory analysis enables the determination of consumer preferences and acceptability for a specific food product, providing valuable information for product development and improvement. Descriptive evaluations are also employed, wherein panelists (consumers) describe products using terms based on sensory attributes.”
- L54-56. In this sentence you are mixing descriptive techniques that must be performed with panels of highly trained judges with techniques that can be performed with semi-trained judges or consumers. There is no discussion of rapid descriptive methods or why they were developed, or the advantages they have over classical descriptive methods. More information should be given about the flash profile technique.
Response: Thanks for your comment and, after conducting a thorough review of our manuscript, we recognize that it lacked depth in comparing rapid descriptive methods with conventional descriptive methods. In the discussion section, we primarily focused on the obtained results and overlooked the aspect you mentioned. In response to your observation, we are pleased to provide the following information you suggested adding to enhance the quality of the study, in lines 103-108:
“Among the descriptive evaluations is the Flash Profile, which is a rapid and flexible method of sensory analysis where the panelists (participants) do not require training because they can use words from their own vocabulary to describe the product they are evaluating [8]. The purpose of this sensory analysis is to sensorially characterize the products, classify them according to the preferences and sensory attributes indicated by consumers and collect information for the development of new products [9]”
- L67-70. If the objective was only to characterize 4 vegetarian or vegan sausages from a physicochemical and sensory point of view, the work does not present anything new. If the goal was to design an ovo-vegetarian sausage and compare it with others on the market, it could be more interesting. I suggest rewriting the objective and the paper.
Response: Done. Thank you for your comment. Indeed, the objective of the study was to develop a new product and compare its physicochemical and sensory characteristics with products already available in the local market, as indicated in the updated abstract. We improved the aim of our research including more accurate information in lines 115-121, with the following information:
“The objective of the study was to determine if the developed Ovo-vegetarian Sausage (referred to as SO, with 15% chickpea flour, 51% albumin, and 34% soy protein concentrate) exhibited improved physicochemical and sensory characteristics compared to the vegetarian sausages already present in the local market. These sausages, which have been accepted and available for several years, corresponding to: the Classic Vegan Sausage SC, the Vegan Fine Herb Sausage SH, and the Quinoa Sausage SQ”
- L73-79. The preparation of the ovo-vegetarian sausage is poorly explained. How was that formulation developed? The most interesting part of the work could be the development of this product and its comparison with others on the market. Was it considered when buying sausages on the market that they had a similar shelf life?
Response: We value your observations as they are essential for enhancing the quality of the study, and we appreciate the suggestions you have provided. In terms of formulating the product, preliminary tests were conducted to determine the optimal proportions of chickpea flour, soy protein concentrate, and albumin, which were found to be 15%, 34%, and 51%, respectively. Additionally, we defined the quantities of other protein sources (soy protein isolate and wheat gluten), vegetable oil, flavor-enhancing ingredients (salt, garlic powder, onion powder, oregano, and sausage flavoring), and colorings (beet extract and carmine). As for the preparation process, it proceeded as follows (improved in lines 125-135):
“The ingredients, including egg albumin, soy protein concentrate, chickpea flour, soy protein isolate, and gluten, were initially hydrated by soaking them in cold water (3°C). Subsequently, the hydrated ingredients were blended using a food processor. The remaining ingredients were then gradually added until a uniform mixture was achieved. Oil was incorporated to form an emulsion. The mixture was then filled into synthetic casings measuring 20 mm in diameter and 12 cm in length, which were manually twisted and tied. The uncooked sausages were wrapped in polyethylene cling film and heated at 80°C until they reached an internal temperature of 72°C. After cooking, the sausages were rap-idly cooled to 18°C using water and stored at 4°C for 24 hours until the corresponding analyses were conducted. Furthermore, measures were taken to ensure that sausage samples purchased from the local market had the same shelf life.”
- L102-105. There is a lack of bibliographic references and information on consumer recruitment. The term “untrained panelists” would seem to be synonymous with consumer, but this is not clear and leads to confusion for the reader. It would also seem that there were two recruitments: one of individuals to carry out the flash profile methodology and another to evaluate acceptability and purchase intention of the samples, but this is not well explained. The number of consumers to evaluate acceptability and purchase intention is very low. Were the recruited individuals regular consumers of vegetarian products? This information is very important. Why wasn't a quick descriptive methodology carried out with the same consumers who evaluated acceptability, such as CATA or RATA? The results could have been more reliable.
Response: We appreciate your comment and have taken it into consideration to ensure clarity for readers. Therefore, we have revised the term "panelists" to "consumers."
Indeed, two phases were added in the improved manuscript (lines 222 – 229). For the Flash Profile analysis, we worked with a group of 15 consumers, following the recommendation of Varela & Gaston (2014) who suggest working with 15–20 consumers for rapid descriptive tests. Similarly, Liu et al. (2018) worked with groups of 10 consumers for this type of analysis. For evaluations of overall liking and purchase intention, we engaged with a group of 60 consumers in line with the UNE-EN ISO 11136:2017 standard, which recommends a minimum of 60 participants. Additionally, overall liking results have been reported with 50 consumers in another research (Hiscock et al., 2018). This modification has been implemented in the manuscript based on your valuable suggestions. You can find additional information on how the sensory analysis was conducted in the updated version of the article that is being sent to you. Furthermore, the consumers recruited for the sensory analysis were individuals following a vegetarian diet. The CATA or RATA methodology was not employed due to budget constraints within the project. Moreover, the bibliographic references added in the manuscript are:
- Varela, P.; & Gaston Ares, eds. Novel techniques in sensory characterization and consumer profiling. CRC Press, 2014.
- UNE - EN ISO 11136:2017/A1:2021. Análisis sensorial. Metodología. Guía general para la realización de pruebas hedónicas con consumidores en una zona controlada. 2021, ed. AENOR.
- Hiscock, L.; Bothma, C.; Hugo, A.; Van Biljon, A.; & Jansen Van Rensburg, W. S. Overall liking and sensory profiling of boiled Amaranthus leaves using the Check-all-that-apply question. CyTA - Journal of Food 2018, 16(1), 822–830. doi: 10.1080/19476337.2018.1464521
- L112-114. What were the terms that were agreed upon?
Response: The word “term” refers to the descriptors and has been indicated in the abstract.
- L117-122. There is a lot of information missing about how this test was performed. Where was the study carried out? Were the samples presented at random? What was the serving temperature? Was there any prior preparation of the samples?
Response: We appreciate your remark, and we would like to provide the following information: The sensory analysis was conducted at the Sensory Analysis Laboratory of “Universidad Peruana Unión (UPeU)”. A flash profile analysis, consisting of two parts, was conducted. In the first stage, consumers tasted the four sausage samples (each 1 cm long and, conditioned at 25°C) and indicated the sensory attributes using terms from their own vocabulary. In the second stage, they ranked the sausage samples in increasing order on a line according to each indicated attribute. For this test, 15 students from UPeU were gathered, ranging in age from 18 to 30 years, with a gender distribution of 40% male and 60% female. Similar to the previous evaluation, each participant assessed the samples in a closed space with good lighting and hygienic conditions. The samples were presented on disposable plates with a different random code for each one and were accompanied by a glass of water for participants to drink after trying each sample.”
On the other hand, a total of 60 participants, ranging in age from 18 to 30 years, including students and campus staff, were assembled to form the untrained consumer panel for overall liking and purchase intention evaluation. The overall liking of the optimal ovo-vegetarian sausage and three commercial vegan sausages (classic vegan sausage, vegan sausage quinoa, and vegan sausage with fine herbs) was assessed on a 9-point hedonic scale (1: I dislike it very much, 9: I like it very much) and purchase intention using a scale from 1 to 5 (1: I would not buy it, 5: I would buy it). Each participant evaluated the samples in a closed space with good lighting and hygienic conditions. The samples, cut to 1 cm in length, were conditioned at room temperature (25°C) and presented on disposable plates with a different random code for each one. Additionally, each sample was accompanied by a glass of water for participants to drink after tasting. All this information is in the improved manuscript, in lines from 204-215.
- ANOVA on flash profile data? This is a mistake.
Response: We appreciate your observation, and yes, it was a mistake. The right term is “PANOVA” (two phase-only analysis of variance), and was corrected in the improved manuscript (line 270).
- L245-259. The sample with the highest overall liking did not reach the minimum commercial acceptability (6.0 on a 9-point scale). Could it have been because the consumers were not vegetarians? Everything is very poorly explained and there is no good discussion of the results. If a larger number of consumers had been used for the study, groups with different preference patterns could have been found, which would have enriched the results.
Response: The analysis was conducted with students who adhere to a vegetarian diet and are enrolled in a university that offers a Master's Degree in Human Nutrition with a specialization in Vegetarian Nutrition. This suggests that there may be vegetarians among their student population. A summary of this whole idea, was included in the improved manuscript, in lines 350-351.
- L269-L270. A figure is not necessary to present. A table with the average values and their standard deviations would be more useful to the reader. Furthermore, the figure leads to confusion by presenting data from two different scales.
Response: Dear reviewer, first of all, we value your feedback about this point. However, we believe that due to the limited data, a figure is more illustrative than a table, which we find too simplistic for presentation in a scientific paper. To address your concern, we have included the average values for general acceptability and purchase intention in the figure.
Round 2
Reviewer 1 Report
Comments and Suggestions for Authors
Authors answered rightly to my comments and added more information as requested, in my opinion the revised version of the manuscript can be acceptable.
Author Response
Thank you for your appreciation reviewer 1.